# Inverse Synthetic Aperture Radar Sparse Imaging Recovery Technique Based on Improved Alternating Direction Method of Multipliers

**DOI:** 10.3390/s25092943

**Published:** 2025-05-07

**Authors:** Hongxing Hao, Wenjie Zhu, Ronghuan Yu, Desheng Liu

**Affiliations:** National Key Laboratory of Space Awareness, Space Engineering University, No.1 Bayi Road, Huairou, Beijing 101400, China; hongxinghao87@hgd.edu.cn (H.H.); yrh1983@163.com (R.Y.); liudesheng@hgd.edu.cn (D.L.)

**Keywords:** inverse synthetic aperture radar, sparse imaging, alternating direction method of multipliers, compressed sensing, orthogonal matching tracking algorithm

## Abstract

Inverse synthetic aperture radar (ISAR) technology is widely used in the field of target recognition. This research addresses the image reconstruction error in sparse imaging for bistatic radar systems. In this paper, sparse imaging technology is studied, and a sparse imaging recovery algorithm based on an improved Alternating Direction Method of Multipliers is proposed. The algorithm accelerates the convergence of the algorithm by dynamically adjusting iterative parameters in the iterative process. Experiments show that the algorithm proposed in this paper has lower relative recovery error in the case of different noise levels and sparsity, and it can be concluded that the algorithm proposed in this paper has a lower relative recovery error than the ADMMs (Alternating Direction Method of Multipliers).

## 1. Introduction

The advancement of radar detection technology has led to the widespread application of inverse synthetic aperture radar sparse imaging recovery technology based on compressed sensing theory in the field. The development of compressed sensing theory, along with progress in solving algorithms [1] and research on regularization methods, has facilitated the extensive study [2] of sparse signal-processing technology in radar detection. The theoretical framework for transforming signals from the time domain to the frequency domain has given rise to the concept of sparse signal recovery. Despite numerous challenges, advancements in technology have led to the development of various solution algorithms. Simultaneously, sparse coding algorithms and the demands of radar signal processing have mutually driven progress, resulting in a series of applications and innovations in the radar field. Against this backdrop, the developmental trajectory is as follows.

Sparse signal recovery involves transforming time domain signals into frequency domain signals with limited bandwidth, utilizing the Fourier transform and cosine transform. The evolution of transform theory, wavelet transform, and compressed sensing theory has enabled sparse recovery technology to surpass the Nyquist sampling law, recovering original signals from observations below the theoretical sampling value. Sparse observation restoration typically involves minimizing a problem with an objective function consisting of restoration error and sparse induced terms. The regularization term, also known as the sparse induction term, primarily enforces sparsity through the L0 norm. However, the non-convex nature of the L0 norm poses challenges in optimization due to its NP-hard problem [3] nature. In order to solve this problem, Mallat et al. [4] applied the matching pursuit algorithm (MP) to the signal sparse decomposition task for the first time, and then the improved orthogonal matching pursuit algorithm [5] (OMP) was applied to this field, in addition to the iterative hard threshold algorithm (IHT) [6] and the hard threshold pursuit algorithm (HTP) [7]. On the other hand, in order to solve the nonconvex optimization problem with the L0 norm, Donoho et al. [8] used the L0 norm solution and proved the equivalence between the minimum L0 norm solution and the minimum L1 norm solution. Some scholars have studied the convex relaxation problem of the L1 norm, which is also known as the LASSO (least absolute shrinkage and selection operator) problem [9]. There are also a series of algorithms to solve this problem, such as the least-angle regression algorithm (LARS) [10], and so on.

The rapid progress of sparse coding algorithms and the increasing need for sparse recovery in radar signal processing highlight the importance of utilizing sparse coding technology to enhance radar imaging performance. Sparse coding technology aids in overcoming challenges in radar signal processing, such as detection and anti-jamming performance, to achieve high-quality imaging [11]. At the same time, the motion of some targets can also lead to the occurrence of sparse phenomenon, about which some scholars have given some solution algorithms [12]. With the development of new radar systems, frequency domain sparse radar, bistatic radar, and so on, have appeared. Among them, the authors [13] point out that due to the bistatic transceiver characteristics and the need to constantly switch beams when observing the target, it is easy to cause echo loss and sparse aperture in the observation process. Various algorithms, such as the ROMP algorithm, have been proposed to improve inverse synthetic aperture radar image resolution [14]. The literature [15] proposes an algorithm to obtain images using Bayesian bistatic ISAR in the case of a low signal-to-noise ratio. Qiu et al. [16] proposed a model that combines low-rank and sparse prior constraints for inverse synthetic aperture radar imaging to address the imaging problem under different sparse data patterns. Furthermore, they developed two reconstruction algorithms within the framework of the alternating direction method of multipliers. Zhang et al. [17] proposed a sparse ISAR autofocus imaging algorithm based on sparse Bayesian learning. In the literature [18], the Cauchy–Newton algorithm was proposed to solve the weighted L1 norm-constrained optimization problem to realize target image reconstruction.

This study aims to address sparse imaging recovery in inverse synthetic aperture radar by establishing a sparse aperture imaging model and proposing an inverse synthetic aperture radar sparse imaging restoration algorithm based on an improved exchange multiplier direction method. The algorithm’s effectiveness is evaluated through analysis of recovery outcomes under varying noise levels and sparsity, demonstrating its efficacy.

The contributions of this paper are concluded as follows:The sparse imaging model of the inverse synthetic aperture radar is transformed into a constrained problem.An improved Alternating Direction Method of Multipliers method with control parameters updated according to the iterations to accelerate the convergence.

## 2. Sparse Aperture Imaging Model

The physical ISAR imaging process is shown as follows. The echo signal $S_{a}$ can be derived from the ISAR image **X** and the echo model matrix $D_{a}$.

The radar echo model used in this paper, as described in the literature [13], assumes that there are *Q* pulses in the full-aperture echo signal with a cumulative rotation angle of Δ*θ*, and the definition of the sparse basis matrix is as follows:(1)$$D_{a}{|}_{Q\times M}=\left(\begin{array}{cccc} 1 & 1 & \cdots & 1\\ 1 & \omega_{1}^{1} & \cdots & \omega_{1}^{M-1}\\ \vdots & \vdots & \ddots & \vdots \\ 1 & \omega_{Q-1}^{1} & \cdots & \omega_{Q-1}^{M-1} \end{array}\right)$$

Among them, *N* is the number of range units in the ISAR image and *M* is the number of Doppler units. *Q* is the number of the valid pulses. $\omega_{q}^{m}=\exp\left(-j2\pi \frac{mN}{M}\frac{\theta (q)}{\Delta \theta }\cos\frac{\beta (q)}{2}\right)$, in which subscript *q* is the index of the pulsed and superscript *m* is the index of the Doppler units. *θ*(n) and *β*(n) are the rotation angle and bistatic angle change over time during the imaging period, respectively. The two-dimensional target is imaged into *N* range units and *M* Doppler units by matrix **D*_a_***, but in practice, the echo signal received by bistatic ISAR is interfered by noise, so the model of the echo signal is(2)$$S_{a}=D_{a}X+\varepsilon_{0}$$

Formula (2) is the observation model of the ISAR image, which describes the echoes from the targets. More details can be acquired from the literature [13]. In Formula (2), **S***_a_* is the full aperture 2D echo data after motion compensation and phase compensation, **X** is the signal to be recovered, and $\varepsilon_{0}$ is the noise in the measurement process. In the actual measurement, it is not guaranteed that all the aperture data can be completely received. Assume that the number of valid pulses that can be received is K(K ≤ Q), then the actual reception of the fusion of the effective data is(3)$$S=I(D_{a})X+\varepsilon =\mathrm{DX}+\varepsilon$$

The column selection function of the matrix is I(⋅). We selected a few columns of the matrix to be operated and set the other columns to 0. In this thesis, the recovery of **X** is mainly carried out through **S** and **D** pairs, and obviously the larger the number of effective pulses ***K***, the better the recovery of the signal is. For the noise ε, in this study, we assume that it is Gaussian white noise [13], The probability density function of the noise is(4)$$P(\varepsilon )={\left(\frac{1}{2\pi \sigma^{2}}\right)}^{\mathrm{NK}}\exp\left(-\frac{1}{2\sigma^{2}}{\left\|\varepsilon \right\|}_{2}^{2}\right)$$
where known **X**(5)$$P(S|X)={\left(\frac{1}{2\pi \sigma^{2}}\right)}^{\mathrm{NK}}\exp\left(-\frac{1}{2\sigma^{2}}{\left\|S-\mathrm{DX}\right\|}_{2}^{2}\right)$$

Since radar signals are assumed to be well sparse in this study, it can be assumed that each pixel point $x_{\mathrm{ij}}=X(i,j)$ in the target image obeys a Laplace distribution with the parameter $l_{\mathrm{ij}}$. Thus,(6)$$P(x_{\mathrm{ij}})=\left(\frac{l_{\mathrm{ij}}}{2}\right)\exp\left(-l_{\mathrm{ij}}{\left\|x_{\mathrm{ij}}\right\|}_{1}\right)$$

Assuming that the individual pixel points follow an independent distribution, the probability density function of **X** is(7)$$P(X)=\left[\prod_{i=1}^{N}\prod_{j=1}^{M}\left(\frac{l_{\mathrm{ij}}}{2}\right)\right]\exp\left(-\sum_{i=1}^{N}\sum_{j=1}^{M}l_{\mathrm{ij}}{\left\|x_{\mathrm{ij}}\right\|}_{1}\right)$$

To estimate **X**, according to the Bayesian criterion,(8)$$\begin{array}{c} \bar{X}=\mathrm{argmax}P(X|S)\\ =\mathrm{argmax}P(S|X)\cdot P(X)\\ =\mathrm{argmax}(\log P(S|X)+\log P(X)) \end{array}$$

Substituting and simplifying the above public notice yields(9)$$\begin{array}{cc} \bar{X} & =\mathrm{argmax}\left\{-\frac{1}{2\sigma^{2}}{\left\|S-\mathrm{DX}\right\|}_{F}^{2}-{\;\left\|L⊗X\right\|}_{1}\right\}\\ & =\mathrm{argmin}\left\{\frac{1}{2}{\left\|S-\mathrm{DX}\right\|}_{F}^{2}+\sigma^{2}{\left\|L⊗X\right\|}_{1}\right\} \end{array}$$
where(10)$$L=\left(\begin{array}{cccc} l_{11} & l_{12} & \cdots & l_{1M}\\ l_{21} & l_{22} & \cdots & l_{2M}\\ \vdots & \vdots & \ddots & \vdots \\ l_{N1} & l_{N2} & \cdots & l_{\mathrm{NM}} \end{array}\right)$$

⊗ is the multiplication of the corresponding elements of the matrix, and ${\left\|U\right\|}_{F}$ is the Frobenious norm for the matrix U, which is defined as ${\left\|U\right\|}_{F}=\sqrt{\sum_{i,j}U(i,j)}$, and U(i,j) is the element with index (i,j).

In the optimization process, the optimization of $\bar{X}$ will be solved according to the columns, and the matrix **L** is called the regularized sparse parameter, which is used to balance the recovery error term and the regularization term. If the noise level is larger, the corresponding element of **L** should take a larger value. If the noise level is smaller, the corresponding element of **L** should take a smaller value. In this study, we assume **L** is **the** same value, then the problem can be further simplified.

## 3. Inverse Synthetic Aperture Radar Sparse Imaging Recovery Based on Improved Alternating Direction Method of Multipliers

In order to solve the above optimization problem, this thesis proposes a solution method based on the improvement of the Alternating Direction Method of Multipliers (ADMMs), which is derived from the traditional one [19], where the above problem is first transformed into a constrained problem.(11)$$\begin{array}{c} \underset{X\in C^{M\times N}}{\min}\left(1/2\right){\left\|S-\mathrm{DX}\right\|}_{F}^{2}+\lambda {\left\|U\right\|}_{1}\\ s.t.\;X=U \end{array}$$

The Lagrange multiplier method is then applied to add quadratic terms X−U22 to the objective function of the above problem. The algorithm does this by alternately minimizing the original variables **X** and **U** and maximizing the dyadic variables (a version of the Lagrange multiplier method). The pseudo-code of the algorithm is given by Algorithm 1.
**Algorithm 1.** Inverse synthetic aperture radar sparse imaging recovery algorithm based on Improved Alternating Direction Method of MultipliersInput: D∈CQ×M, S∈CQ×N, μ>0 (multiplier), λ>0 (noise parameter)Output: X∈CM×N1. **Begin**2.    *k* **= 0**3.    X(0)=DHS 4.    U(0)=X, V(0)=05.    F=(DHD+μI)−16.    **While** not converge **do**7.        U(k+1)=soft(X(k)−V(k),λ/μ)8.        X(k+1)=F(DHS+μ(U(k+1)+V(k)))9.        V(k+1)=V(k)−(X(k+1)−U(k+1))10.      k=k+111.    **end**12.   X=X(k)13. **end**


The complexity of this algorithm to encode N signals using the same matrix is $O(KQ^{2}+\mathrm{NKQ})$. In this algorithm, the function $\mathrm{soft}\left(x,\tau \right)=\max\left\{\left|x\right|-\tau ,0\right\}\left(x/\left|x\right|\right)$ is a soft threshold method and it operates for each element of the matrix. When the value of ***N*** is large, step 7 of the algorithm is the most time-consuming step with a complexity of $O(Q^{2})$. By utilizing the left eigenvector and eigenvalue of the matrix $D^{H}$, the step can be simplified to an operation with a complexity of O(KQ), in which K is the sparsity of the signal representation. That is, we just use the K column of the dictionary D to represent signal S. If a large K is used, the signal will be recovered perfectly, as well as the noise. Otherwise, the signal will not be recovered perfectly, but the noise will be suppressed as well. Although the value of μ>0 affects the final convergence rate, the convergence of the algorithm is independent of its value. The selection of μ is discussed in detail in the literature [10].

In this study, the parameters μ are changed during the iteration process to achieve a faster convergence process, where the focus is on comparing the change of the optimization variable **X** with its dyadic variable **U** and comparing **U** with the result of the number of iterations corresponding to multiples of 10. The following algorithm shows how to update parameter μ depending on the relationship between the two changes.

Our proposal combines Algorithms 1 and 2. The algorithm can effectively accelerate the convergence speed and ensure that the objective function quickly converges to the optimized objective value. The adjustment of control parameter *p,* which is a real number, is selected randomly, taking the value of 10 in this paper. The smaller the *p*, the more frequent the updating of the parameters μ and V.(12)$$\underset{X\in C^{M\times N}}{\min}\left(1/2\right){\left\|S-\mathrm{DX}\right\|}_{F}^{2}+\lambda {\left\|U\right\|}_{1}+\mu {\left\|X-U\right\|}_{2}^{2}$$

In Algorithm 2, the parameter μ is changed according to the relationship between $r_{1}$ and $r_{2}$. If $r_{1}>r_{2}$, which means the different between X and U is larger than between U and $U_{0}$, μ should be increased to add more penalty to the third term of the object function shown in Formula (12), and vice versa. So, the convergence of the method can benefit from the tuning of parameter μ.

Although we change parameters μ and V according to the iterations, the complexity of the algorithm is still not changed, but in each iteration the penalty is changed to make the independent variables U and X optimal.
**Algorithm 2.** Algorithm for updating the parameters of Improved Alternating Direction Method of MultipliersInput: i (number of iterations), X, U, V, μ>0 (multiplier), p>0 (adjustment of control parameter)Output: μ’ (updated multiplier), V’ (iterative variables)1. **Begin**2.    if imod⁡10=03.        U0=U4.    **end**5.    r1=X-UF6.    r2=U-U0F7.     μ’=μ8.     V’=V9.    if r1>pr210.       μ’=2μ 11.       V’=V/212.  **end**13.   if r2>pr114.       μ’=μ/215.       V’=2V16.   **end**17.**end**


## 4. Results and Discussion

The experimental part selects the simulated imaging results of inverse SAR of three different models of airplanes, in which the inverse SAR image sizes of the three models are 64 × 128, 128 × 64, and 128 × 128, respectively, in which the first number is the quantity of the cross range with denotation *M* and the second number is the Doppler range with denotation *N*. The physical quantity of the axes is the number of image pixels D, which is dependent on the measurement matrix with size Q×M. In the following experiments, D is initialized by a random matrix with Q=128. The original *K* of the three models are 64, 128, and 128, respectively. Their original interferometric SAR images, which are notated by **X**, are shown in Figure 1:

First, the speed of the convergence of the improved algorithm of this thesis is compared with respect to the original method of swapping multiplier directions. For different model airplanes selecting a line of measurement, the change in the objective function value with the number of iterations is shown in Figure 2.

Figure 2 shows the convergence curves of model 1–model 3 at the 10th, 30th, and 60th measurement points with equal signal-to-noise ratios, where the horizontal coordinate is the number of iterations and the vertical coordinate is the value of the objective function. Comparison of the results leads to the conclusion that the convergence speed of the improved algorithm is higher than that of the original Alternating Direction Method of Multipliers. The degree of improvement varies from case to case, e.g., for model 2 observation 30, the improvement starts at the 12th iteration step and is small relative to the other two cases.

Figure 3 shows the convergence speed of each algorithm for different noise cases for model 1. It can be seen that the objective function values converge to larger values for the more noisy case (signal-to-noise ratio of 10). Conversely, with less noise, the objective function value converges to a smaller value, as shown in Figure 4. On the other hand, in terms of convergence speed, the case with relatively small noise has a faster convergence speed and is already close to the final convergence value at the 10th iteration, whereas the case with larger noise (signal-to-noise ratio of 10) has a relatively slower convergence speed and is probably close to the final convergence value at the 25th iteration.

In order to quantitatively analyze the effect of sparse recovery, X is the original image and $\bar{X}$ is the processed image. The relative recovery error (RRE) is defined as(13)$$r={\left\|X-\bar{X}\right\|}_{F}/{\left\|X\right\|}_{F}$$
where ${\left\|X\right\|}_{F}$ is the Frobenius paradigm, which is defined as(14)$${\left\|X\right\|}_{F}=\sqrt{\sum_{i}\sum_{j}{\left|x_{\mathrm{ij}}\right|}^{2}}$$

For model 1, the relative errors of recovery at different noise levels and with measurement sparsity are shown in Table 1 below. The imaging results are shown in Figure 5 and Figure 6 below. In this experiment, our proposal achieves the same results as the ADMMs for different SNRs and a sparsity of K, since our proposal is an improvement of the ADMMs, and the worst results are the same as the ADMMs when *p* in Algorithm 2 is very large. For the same SNR, the least RRE can be achieved for the largest sparsity K since the fitting error is small in this setting. Detailed analysis of the effect of the SNR and K on the RRE will be shown in the following experiments.

For model 2, the relative errors of recovery at different noise levels and with measurement sparsity are shown in Table 2 below. The figure marked bold is the best result among all the compared algorithms. The imaging results are shown in Figure 7, Figure 8, Figure 9 and Figure 10 below. In this experiment, our proposal achieves the best results, especially for the experiments with large sparsity K (the last column of Table 2). From the white rectangle in Figure 8 and Figure 9, we can conclude that our proposal suppresses the noise better than the other algorithms.

For model 3, the relative errors of recovery at different noise levels and with measurement sparsity are shown in Table 3 below. The figure marked bold is the best result among all the compared algorithms. The imaging results are shown in Figure 11, Figure 12, Figure 13 and Figure 14 below. In this experiment, the same conclusion can be achieved as plane model 2.

Figure 15 shows a comparison plot of the recovery error for different noise and sparsity values. Figure 15a shows the recovery error for SNR = 5 for model 2 at different sparsity values. Analyzing the left figure, it can be concluded that as the value increases and the sparsity decreases, the recovery error first decreases and then increases, while as the sparsity decreases, the better the fit to the signal and, therefore, the error decreases. However, since the noise is more drastic in the SNR = 5 case, the recovery process recovers the noise as the sparsity continues to decrease, so the recovery error relative to the original signal gradually increases. For the recovery errors calculated by the three algorithms, the OMP algorithm has the worst recovery, and the algorithm proposed in this paper has the best recovery. For Figure 15b, SNR = 50, the noise is relatively small. As a result, the recovery error decreases gradually as the sparsity decreases and more received signals are used to recover the original signal. The algorithm proposed in this paper can obtain a smaller relative recovery error compared to the other two algorithms.

The OMP method is a greedy method that is not definitely convergent to the optimal, so the RRE is large. The ADMMs and our proposal is the convex approximation, and our proposal tunes the parameters to get a better solution to the original constrained problem than ADMMs, so the least RRE can be got.

Figure 16 shows the relative recovery error curves of different algorithms for different signal-to-noise ratios. Figure 16a shows the relative recovery error plot for model 2 with sparsity K = 10. As the noise decreases, the different algorithms stabilize at a certain relative recovery error value, which is due to the error due to sparsity. And among them, the algorithm proposed in this thesis that stabilizes in the relative recovery error is smaller than the other two algorithms, and the algorithm in this thesis is able to achieve better sparse recovery results. Figure 16b shows the relative recovery error for model 3 in the sparsity K = 5 case, which stabilizes at a certain value as the noise decreases, and again, the algorithm proposed in this thesis is able to obtain a smaller relative recovery error.

Figure 17 shows, for model 2, the relative recovery error curves for different λ values taken at different noises in the sparsity K = 20 case. Analyzing the figure below, it can be concluded that the optimal λ value is 5000 in the case of relatively large noise, and the smaller relative recovery error is achieved by increasing the sparsity to suppress the noise. With decreasing noise levels, the optimal “λ” value also decreases. When the signal-to-noise ratio exceeds 15, a “λ” value ranging from 1 to 100 yields a near-optimal relative recovery error.

As shown in Equation (11), the parameter λ balances the fitting error and noise suppression. If the SNR is large, such as 30 dB, a small value of λ should be selected to give a large weight to the fitting error in the objection function; thus, a small fitting error can be achieved by minimizing the objection function. Otherwise, the large value of λ should be used for a small SNR since the large value of λ can suppress the severe noise.

The effect of the proposed algorithm is analyzed below for different parameter update frequencies, where the number i denotes that the parameters are updated every i iterations. Calculate the recovery time for the model 2, K = 10 case. Analyzing Table 4, it can be concluded that the minimum time is consumed to change the parameters every 10 iterations. As i increases, the frequency of parameter changes decreases, the amount of computation decreases gradually, while the speed of convergence also decreases, and there exists an optimal value of i that minimizes the time consumed. In this experiment, the optimal number of iterations is 10.

## 5. Conclusions

This study focuses on addressing the issue of sparse imaging in the context of inverse synthetic aperture radar (ISAR). Initially, an examination of the sparse aperture imaging model is conducted, followed by the introduction of an ISAR sparse imaging recovery algorithm utilizing the Improved Alternating Direction Method of Multipliers (ADMMs). The algorithm’s convergence is enhanced through the adjustment of iterative parameters. Experimental results demonstrate the algorithm’s capability to achieve recovery outcomes with significantly reduced relative recovery error. Furthermore, the thesis evaluates the impact of noise and sparsity on the recovery process and conducts a sensitivity analysis of the algorithm’s parameters.

## Figures and Tables

**Figure 1 sensors-25-02943-f001:**
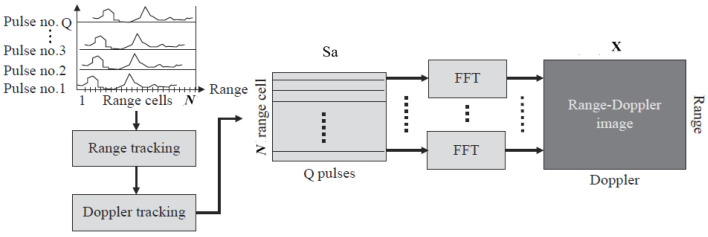
The ISAR imaging system [20].

**Figure 2 sensors-25-02943-f002:**
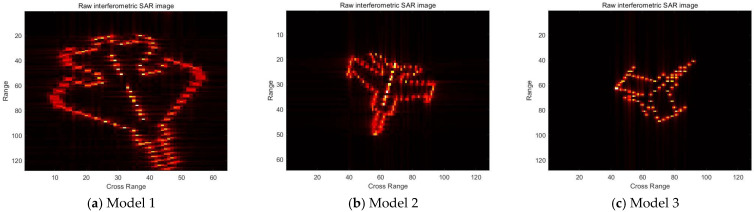
The simulated imaging results of inverse SAR of three different models of airplanes.

**Figure 3 sensors-25-02943-f003:**
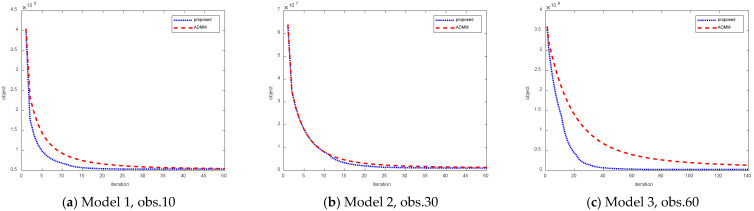
Variation of objective function values at different observation points for different models.

**Figure 4 sensors-25-02943-f004:**
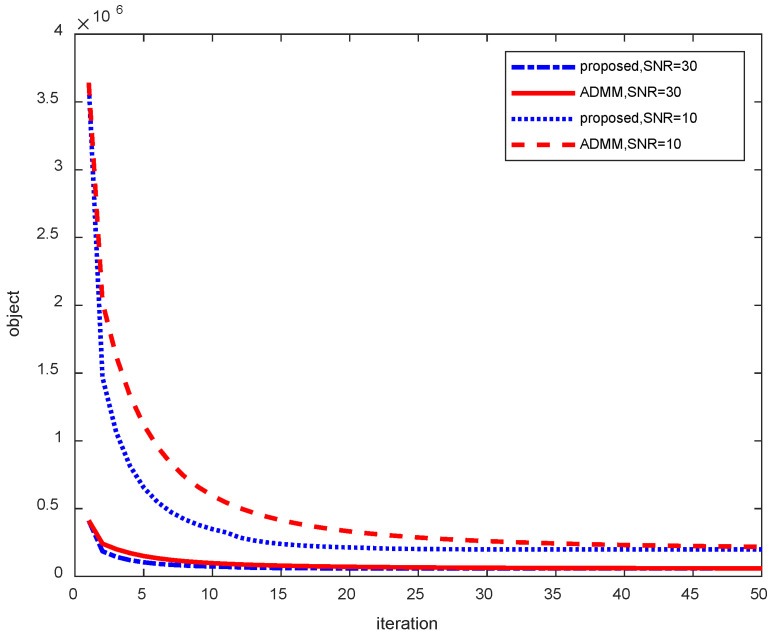
Convergence rate under varying noise conditions.

**Figure 5 sensors-25-02943-f005:**
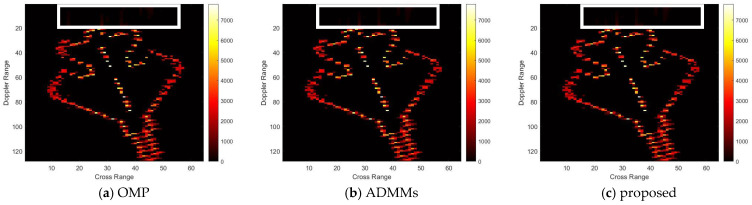
SNR = 50, K = 5 recovery effect for model 1.

**Figure 6 sensors-25-02943-f006:**

SNR = 10, K = 5 recovery effect for model 1.

**Figure 7 sensors-25-02943-f007:**

SNR = 50, K = 5 recovery effect for model 2.

**Figure 8 sensors-25-02943-f008:**
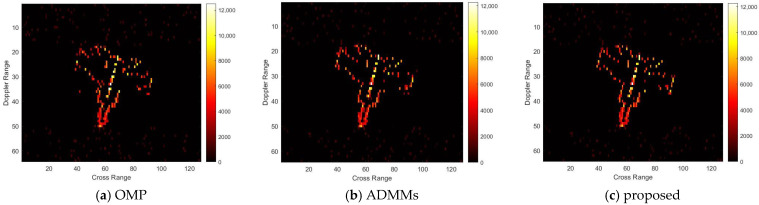
SNR = 10, K = 5 recovery effect for model 2.

**Figure 9 sensors-25-02943-f009:**
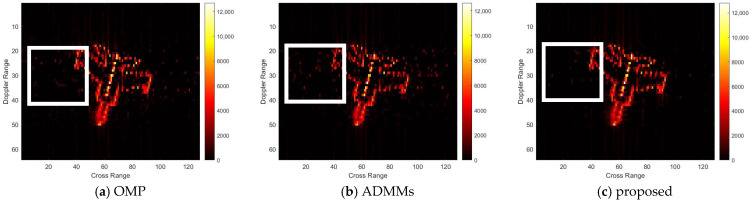
SNR = 50, K = 20 recovery effect for model 2.

**Figure 10 sensors-25-02943-f010:**
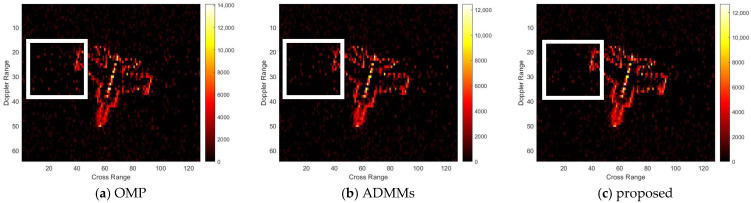
SNR = 10, K = 20 recovery effect for model 2.

**Figure 11 sensors-25-02943-f011:**

SNR = 50, K = 5 recovery effect for model 3.

**Figure 12 sensors-25-02943-f012:**

SNR = 10, K = 5 recovery effect for model 3.

**Figure 13 sensors-25-02943-f013:**
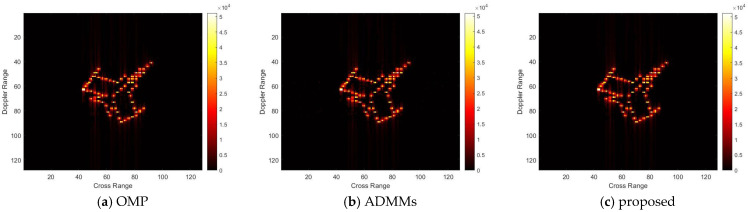
SNR = 50, K = 20 recovery effect for model 3.

**Figure 14 sensors-25-02943-f014:**

SNR = 10, K = 20 recovery effect for model 3.

**Figure 15 sensors-25-02943-f015:**
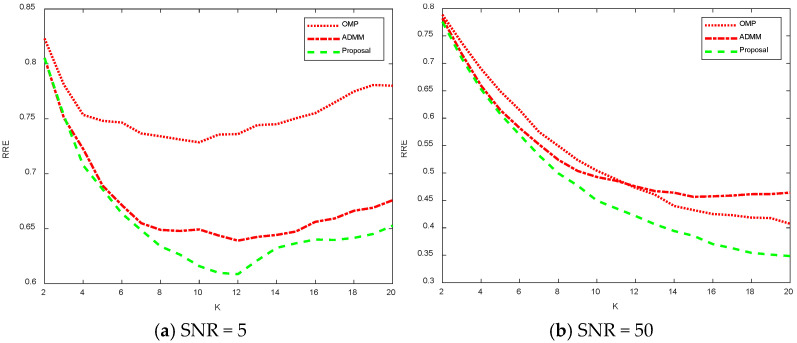
Comparison results of recovery errors for model 2 with different noise and sparsity.

**Figure 16 sensors-25-02943-f016:**
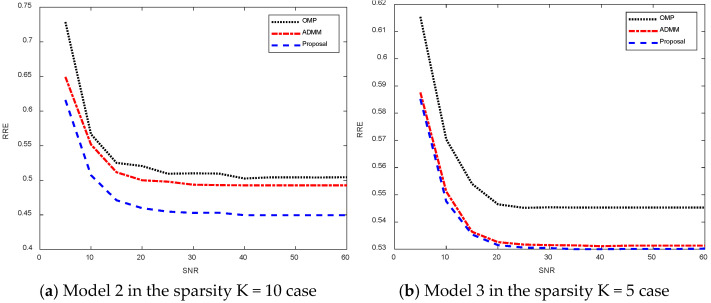
Relative recovery error plots for different signal-to-noise ratios.

**Figure 17 sensors-25-02943-f017:**
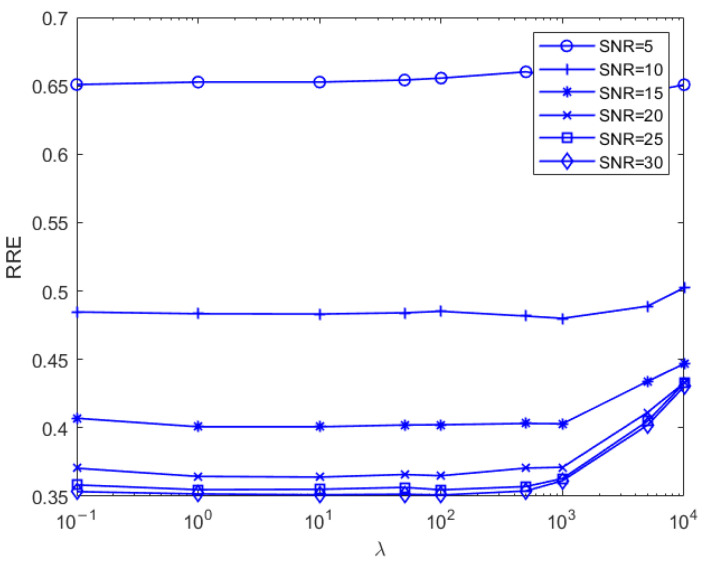
Model 2, relative recovery error plots for different λ values at different noise levels in the sparsity K = 20 case.

**Table 1 sensors-25-02943-t001:** Comparison of relative recovery errors for different noise and sparsity scenarios (model 1).

Noises	Methods	K = 3	K = 5	K = 10	K = 20
SNR = 80	OMP	0.6339	0.5103	0.3426	0.2066
	ADMMs	0.6225	0.4981	0.3287	0.1942
	proposed	0.6225	0.4981	0.3287	0.1942
SNR = 50	OMP	0.6333	0.5103	0.3426	0.2063
	ADMMs	0.6226	0.4981	0.3287	0.1943
	proposed	0.6226	0.4981	0.3287	0.1943
SNR = 30	OMP	0.6333	0.5107	0.3427	0.2085
	ADMMs	0.6227	0.4984	0.3293	0.1956
	proposed	0.6227	0.4984	0.3293	0.1956
SNR = 10	OMP	0.6490	0.5330	0.4037	0.3564
	ADMMs	0.6328	0.5162	0.3813	0.3256
	proposed	0.6328	0.5162	0.3813	0.3256

**Table 2 sensors-25-02943-t002:** Comparison of relative recovery errors for different noise and sparsity scenarios (model 2).

Noises	Methods	K = 3	K = 5	K = 10	K = 20
SNR = 80	OMP	0.7375	0.6480	0.5044	0.4071
	ADMMs	0.7173	0.6148	0.4936	0.4642
	proposed	**0.7098**	**0.6078**	**0.4537**	**0.3498**
SNR = 50	OMP	0.7375	0.6486	0.5042	0.4075
	ADMMs	0.7173	0.6149	0.4936	0.4615
	proposed	**0.7098**	**0.6075**	**0.4537**	**0.3502**
SNR = 30	OMP	0.7377	0.6485	0.5100	0.4090
	ADMMs	0.7176	0.6139	0.4921	0.4596
	proposed	**0.7078**	**0.6073**	**0.4541**	**0.3509**
SNR = 10	OMP	0.7539	0.6827	0.5667	0.5592
	ADMMs	0.7277	0.6363	0.5531	0.5378
	proposed	**0.7237**	**0.6268**	**0.5050**	**0.4852**

**Table 3 sensors-25-02943-t003:** Comparison of relative recovery errors for different noise and sparsity scenarios (model 3).

Noises	Methods	K = 3	K = 5	K = 10	K = 20
SNR = 80	OMP	0.6736	0.5453	0.3119	0.1524
	ADMMs	0.6692	0.5313	0.3018	0.1711
	proposed	**0.6672**	**0.5302**	**0.2983**	**0.1449**
SNR = 50	OMP	0.6736	0.5453	0.3119	0.1524
	ADMMs	0.6692	0.5313	0.3018	0.1711
	proposed	**0.6672**	**0.5301**	**0.2984**	**0.1449**
SNR = 30	OMP	0.6739	0.5454	0.3118	0.1535
	ADMMs	0.6693	0.5315	0.3022	0.1727
	proposed	**0.6673**	**0.5304**	**0.2990**	**0.1463**
SNR = 10	OMP	0.6877	0.5704	0.3763	0.3167
	ADMMs	0.6789	0.5512	0.3544	0.2948
	proposed	**0.6774**	**0.5476**	**0.3527**	**0.2815**

**Table 4 sensors-25-02943-t004:** Parameter update frequency and consumption schedule.

i	2	5	10	20
t	1.3550	1.3194	1.1585	1.4594

## Data Availability

The data presented in this study are available upon request from the corresponding author. The data are not publicly available due to the request of the funder.

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
