# Peer review of "Inverse Synthetic Aperture Radar Sparse Imaging Recovery Technique Based on Improved Alternating Direction Method of Multipliers"

_sensors, 2025, doi:10.3390/s25092943_

Round 1
Reviewer 1 Report
Comments and Suggestions for Authors
This paper proposes an improved ADMM framework for sparse reconstruction optimization in ISAR imaging. According to the author, this approach enhances the convergence efficiency of the algorithm. However, the paper would benefit from standardization in terms of English writing proficiency and paper formatting. Additionally, the paper writing method could be further optimized, and the presentation logic of the innovative work could be further refined. It is recommended that the author invite a tutor or expert in the relevant field to conduct a basic review and check, and the paper is recommended to be major revised OR resubmitted.
- There is considerable scope for enhancement with respect to the quality of the English writing in the paper, with some general and specialized terms demonstrating evident errors in both translation and writing. Additionally, there is a lack of standardization in the capitalization of titles, with significant inaccuracies observed in the writing of pivotal terms. For instance, OMP is erroneously written as OPM, and ADMM is written as ADMN in Figures 4-15. Some mathematical and technical terms are not written with the requisite accuracy, for example, l0 norm should be written as L0 norm or $L_0$ norm, and so on. The citation format is not consistent, and abbreviations of terms should be provided at the first instance of their appearance, such as ADMM, and the abbreviations of the terms should be explained in the text, such as ISAR, and so on. Furthermore, some variables in the formula are not defined, such as $l_ij$.
- The format of the charts is not standardized, as evidenced by disparities in picture size, horizontal and vertical coordinate systems, and font style. The format of Tables 1-3 presents challenges, and the imaging area lacks unification.
- The performance improvement of the proposed methodology should be quantified in the abstract, with the abstract itself making it clear that the radar system under study is a bistatic radar.
- The introduction section is inadequate in its summary of the current research status, and there is a paucity of literature on the application of sparse reconstruction theory in related fields and ADMM-related optimization algorithms to ISAR imaging. The introduction should provide a necessary review of existing research, indicate the problems to be solved in existing research, and list the innovations of this research.
- It is imperative to ascertain whether the improved ADMM described in the text constitutes an existing algorithm. In the event that it is, references should be provided. If it is original to this article, the innovative points should be explained, and the specific steps of the improvement clearly indicated. The method of selecting $p$ in Algorithm 2 and its value are not given; therefore, a description should be added.
If the convergence speed of the algorithm can be improved by changing the value of $\mu$, the logic of the optimization should be explained. A comparison of the proposed algorithm and the OMP algorithm reveals that the former is conducive to enhanced convergence speed. It is imperative to ascertain whether the solution speed has been elevated. A comparison of the computational complexity of the algorithm before and after the enhancement is necessary. The sparsity $K$ definition is not clear in the paper, and the meaning of the author's comparative experiments under different sparsity conditions cannot be understood. A color bar is suggested to be included in the imaging results, and the diagram should indicate the points or areas where the algorithm significantly improves the imaging quality. - The variables in the algorithm flowchart should be given in iterative form.

There is considerable scope for enhancement with respect to the quality of the English writing in the paper, with some general and specialized terms demonstrating evident errors in both translation and writing. The paper would benefit from standardization in terms of English writing proficiency and paper formatting.
Author Response
Reviewer 1:
This paper proposes an improved ADMM framework for sparse reconstruction optimization in ISAR imaging. According to the author, this approach enhances the convergence efficiency of the algorithm. However, the paper would benefit from standardization in terms of English writing proficiency and paper formatting. Additionally, the paper writing method could be further optimized, and the presentation logic of the innovative work could be further refined. It is recommended that the author invite a tutor or expert in the relevant field to conduct a basic review and check, and the paper is recommended to be major revised OR resubmitted.
Respond: Thanks very much for the suggestions. Firstly, we read through the whole manuscript and improve the English writing, and secondly, we modify the paper according to the four experts’ comments and thanks again for all the comments.
Comment1: There is considerable scope for enhancement with respect to the quality of the English writing in the paper, with some general and specialized terms demonstrating evident errors in both translation and writing. Additionally, there is a lack of standardization in the capitalization of titles, with significant inaccuracies observed in the writing of pivotal terms. For instance, OMP is erroneously written as OPM, and ADMM is written as ADMN in Figures 4-15. Some mathematical and technical terms are not written with the requisite accuracy, for example, l0 norm should be written as L0 norm or L_0 norm, and so on. The citation format is not consistent, and abbreviations of terms should be provided at the first instance of their appearance, such as ADMM, and the abbreviations of the terms should be explained in the text, such as ISAR, and so on. Furthermore, some variables in the formula are not defined, such as l_ij.
Respond1: We are very sorry for the mistake. The typographical errors have been corrected as requested (lines 41, 42, 48, 49, 50, 51, 69, 234, 237, 240, 249, 250, 253, 256, 259, 265, 267, 270, 273, and 276). Furthermore, the definition of lij has been provided (lines 106-119 and 107-120).
Comment2: The format of the charts is not standardized, as evidenced by disparities in picture size, horizontal and vertical coordinate systems, and font style. The format of Tables 1-3 presents challenges, and the imaging area lacks unification.
Respond2: Thanks very much for the professional comment. The figures have been revised to incorporate a color bar and the addition of Doppler-range labeling on the y-axis. All figures are now formatted consistently according to the specified template (lines 241, 244, 254, 257, 260, 263, 271, 274, 277, and 280). Furthermore, the formatting of Tables 1 through 3 has been confirmed to adhere to the template (lines 238, 252, and 269). The imaging area is determined by the original radar echo data.
Comment3: The performance improvement of the proposed methodology should be quantified in the abstract, with the abstract itself making it clear that the radar system under study is a bistatic radar.
Respond3: Thanks very much for the professional comment. We have already specified the investigated radar system in the abstract (line 10), and the enhanced performance is demonstrated in the experimental results section.
Comment4: The introduction section is inadequate in its summary of the current research status, and there is a paucity of literature on the application of sparse reconstruction theory in related fields and ADMM-related optimization algorithms to ISAR imaging. The introduction should provide a necessary review of existing research, indicate the problems to be solved in existing research, and list the innovations of this research.
Respond4: Thanks very much for the professional comment. We have incorporated a summary of the paper's contributions at the conclusion of the introduction (line 80). Furthermore, we have included additional references [16] (line 67) that address advancements in ADMM algorithms for sparse aperture ISAR imaging. We have also added two new citations to clarify the existing ADMM algorithm implementations (lines 89, 147).
Comment5: It is imperative to ascertain whether the improved ADMM described in the text constitutes an existing algorithm. In the event that it is, references should be provided. If it is original to this article, the innovative points should be explained, and the specific steps of the improvement clearly indicated. The method of selecting p in Algorithm 2 and its value are not given; therefore, a description should be added.
If the convergence speed of the algorithm can be improved by changing the value of μ, the logic of the optimization should be explained. A comparison of the proposed algorithm and the OMP algorithm reveals that the former is conducive to enhanced convergence speed. It is imperative to ascertain whether the solution speed has been elevated. A comparison of the computational complexity of the algorithm before and after the enhancement is necessary. The sparsity K definition is not clear in the paper, and the meaning of the author's comparative experiments under different sparsity conditions cannot be understood. A color bar is suggested to be included in the imaging results, and the diagram should indicate the points or areas where the algorithm significantly improves the imaging quality.
Respond5: Thanks very much for the professional comment. We introduce the traditional ADMM method and add the reference [18] to the revised version. The contributions of the paper are concluded in the last paragraph of the Introduction section. The method of selecting p in Algorithm 2 and its value are given below Algorithm 2(line 175 to 177). The logic of the optimization by changing the value of μ is also changed below Algorithm 2(line 179 to 183). The computational complexity of the algorithm is analyzed in the revised paper (line 184 to 186). The sparsity K definition is indicated (line 161 to 164).
Comment6: The variables in the algorithm flowchart should be given in iterative form.
Respond6: We change the variables to the iterative form in Algorithm 1(line 155). Thanks for the comment.

Reviewer 2 Report
Comments and Suggestions for Authors
In this work, the authors proposed a sparse imaging recovery algorithm based on improved alternating direction method of multipliers. The algorithm accelerates the convergence of the algorithm by dynamically adjusting iterative parameters in the iterative process.The propsed algorithm has lower relative recovery error in the case of different noise levels and sparsity and a better recovery effect. However, they authors must clarify the following points before its publication:
- The writing of the manuscript is not careful. There are plenty of grammar errors such as in line 15, line 84 (no subject), line 85 (in this thesis?), line 113, line 137 “U as well as U”, line 139, line 142, line 158, line 169, line 178, line 263 ... The authors are required to carefully revise their manuscript’s language issue.
- The titles of Table 1 and 2 are problematic.
- No article before Table1.., or Figure 1....
- The paragraph of line 268-274 repeats that of line 258-264
- The authors only describe the curves that the differences between OMP, ADMM, and proposal algorithms. But they do not explain why?
Comments on the Quality of English Language
English need to be improved greatly.
Author Response
Reviewer 2:
In this work, the authors proposed a sparse imaging recovery algorithm based on improved alternating direction method of multipliers. The algorithm accelerates the convergence of the algorithm by dynamically adjusting iterative parameters in the iterative process. The proposed algorithm has lower relative recovery error in the case of different noise levels and sparsity and a better recovery effect. However, they authors must clarify the following points before its publication.
Respond: Thanks very much for the detailed comments. We take into account the whole comments than revise the manuscript accordingly. The details are shown as follows.
Comment1: The writing of the manuscript is not careful. There are plenty of grammar errors such as in line 15, line 84 (no subject), line 85 (in this thesis?), line 113, line 137 “U as well as U”, line 139, line 142, line 158, line 169, line 178, line 263 ... The authors are required to carefully revise their manuscript’s language issue.
Respond1: We are very sorry for the mistake. All corrections have been implemented (Lines 16, 110, 139, 170, 171, 175, 206, 216, 225, and 320).
Comment2: The titles of Table 1 and 2 are problematic.
Respond2: Thanks very much for the detailed comments. Specific annotations have been incorporated into the model representation, as indicated in lines 238, 252, and 269 of the table.
Comment3: No article before Table1.., or Figure 1....
Respond3: We do not catch the comments exactly. The paper is formatted according to the template. The blank lines before Table1.., or Figure 1.... are determined by the templates.
Comment4: The paragraph of line 268-274 repeats that of line 258-264.
Respond4: We are very sorry for the mistake. The redundant section has been removed.
Comment5: The authors only describe the curves that the differences between OMP, ADMM, and proposal algorithms. But they do not explain why?
Respond5: We give explanations to the differences between OMP, ADMM, and proposal algorithms to get different curves (line 299 to 302).

Reviewer 3 Report
Comments and Suggestions for Authors
The authors developed an improved Alternating Direction Method of Multipliers (ADMM) algorithm for inverse synthetic aperture radar sparse imaging recovery that dynamically adjusts iterative parameters during processing. Their experimental results demonstrate that this improved algorithm achieves lower relative recovery errors compared to traditional methods when tested across different noise levels and sparsity conditions. The following points should be addressed carefully.
- The abstract doesn't clearly articulate what specific problem with existing ISAR sparse imaging technology the research is addressing.
- The introduction jumps between topics without a clear logical flow, making it difficult to follow the progression of ideas from radar detection to sparse imaging challenges. Additionally, there is redundant information, several concepts are repeated across paragraphs without adding new insights.
- Several variables (such as N, M, Q, β(q)) are introduced without clear definitions or contextual explanations. For example, in equation (1), ω's subscripts and superscripts are confusing without proper introduction.
- The connection between equations (1) and (2) isn't clearly explained, leaving gaps in understanding how the sparse basis matrix relates to the full aperture echo data.
- Section 2 (Sparse Aperture Imaging Model) focuses on mathematical formalism without connecting back to the physical ISAR imaging process, making it difficult for readers to understand the practical significance.
- While OMP, ADMMO, and ADMMI are compared, there's an inadequate explanation of what ADMMO represents versus the proposed ADMMI.
- Some figures (like Figure 3) receive detailed analysis while others (like Figures 4-15) receive minimal or no interpretation. please see this !
- Figure 18 shows parameter sensitivity to λ values, the analysis is superficial without exploring the theoretical reasons behind the observed behavior.
- There are grammar mistakes and relatively complicated sentences that need to be addressed, please.
- English quality needs to improve, please
Author Response
Reviewer 3:
The authors developed an improved Alternating Direction Method of Multipliers (ADMM) algorithm for inverse synthetic aperture radar sparse imaging recovery that dynamically adjusts iterative parameters during processing. Their experimental results demonstrate that this improved algorithm achieves lower relative recovery errors compared to traditional methods when tested across different noise levels and sparsity conditions. The following points should be addressed carefully.
Respond: Thanks for the detailed comments and we study through all the comments and revised our paper accordingly. Thank you very much again for helping us in modifying the paper.
Comment1: The abstract doesn't clearly articulate what specific problem with existing ISAR sparse imaging technology the research is addressing.
Respond1: Thanks very much for the comments. The abstract indicates that this paper addresses the image reconstruction error problem in sparse imaging for bistatic radar systems (Line 10).
Comment2: The introduction jumps between topics without a clear logical flow, making it difficult to follow the progression of ideas from radar detection to sparse imaging challenges. Additionally, there is redundant information, several concepts are repeated across paragraphs without adding new insights.
Respond2: Thanks very for the detailed suggestions. We have revised the manuscript extensively, removing redundant information and providing a concise summary in the introduction, specifically at line 27. Furthermore, we have included a summary of the paper's contributions at the end of the introduction, at line 69
Comment3: Several variables (such as N, M, Q, β(q)) are introduced without clear definitions or contextual explanations. For example, in equation (1), ω's subscripts and superscripts are confusing without proper introduction.
Respond3: Thanks very much for the comments. In the modified version of our manuscript, we define the variables and the subscripts and superscripts of the variable ω(line 94 to line 98).
Comment4: The connection between equations (1) and (2) isn't clearly explained, leaving gaps in understanding how the sparse basis matrix relates to the full aperture echo data.
Respond4: Thanks very for the detailed suggestions. We give a detailed description of formula (2)(line 102-line 103). And more details about how the sparse basis matrix relates to the full aperture echo data is described in formula (3). Thanks again for the time in helping us to improve our paper.
Comment5: Section 2 (Sparse Aperture Imaging Model) focuses on mathematical formalism without connecting back to the physical ISAR imaging process, making it difficult for readers to understand the practical significance.
Respond5: Thanks very much for the advice. The physical ISAR imaging process is added to the revised version.(line 86-line 89)
Comment6: While OMP, ADMMO, and ADMMI are compared, there's an inadequate explanation of what ADMMO represents versus the proposed ADMMI.
Respond6: We are sorry for the confusion and thanks very much for the detailed comments. Actually, ADMMo is the original ADMM method and ADMMI is our proposal. In the revised paper, we unified “ADMM method” and “ADMMo method” as “ADMM method”, and unified “ADMMI method” and “proposal method” as “proposal method”. Thanks very much again for pointing out the confusion.(Figure 4 – Figure 15, Table 1 – Table 3)
Comment7: Some figures (like Figure 3) receive detailed analysis while others (like Figures 4-15) receive minimal or no interpretation. please see this !
Respond7: Thanks very much for the comments, and we give interpretation to Figures 4-15. (line 240-line 245, line 256-line 259, line 275-line 276)
Comment8: Figure 18 shows parameter sensitivity to λ values, the analysis is superficial without exploring the theoretical reasons behind the observed behavior.
Respond8: Thanks very much for this comments. We give theoretical reasons behind the observed behavior in the revised paper(line 326-line 330).
Comment9: There are grammar mistakes and relatively complicated sentences that need to be addressed, please.
Respond9: We check the whole paper and correct some typos in the manuscript. Thanks very much again for the comments.(line 120, line 311, line 319 and so on)

Reviewer 4 Report
Comments and Suggestions for Authors
The paper is devoted to the actual problem of reducing the time of registration of bistatic radar echoes when operating in sparse mode, but with image acquisition that could be obtained by careful scanning of space.
I recommend a figure at the very beginning of Section 2 with a brief explanation of the principle of operation and an indication of the variables used.
It is necessary to explain why the noise in the image pixels is Laplace distributed rather than Rayleigh distributed, which is typical for images. At least optical ones. Or make a proper reference!
In formula (11), what is F F=2?Or does the explanation only appear in formula (13)? And how does formula (13) differ from the usual l2 norm?
In section 4, we need to establish a correspondence between the numbers 34 and 128 and the variables M, Q, N and K. How were the images in Figure 1 modeled. Is this image X? What is the value of K for these images?
In Fig. 1, the numbers of samples are plotted on the axes. We need to specify the physical quantities on both axes as well.
I don't understand what “three measurement positions of ground 10, 30, and 60” means. I need an explanation.
“ADMMO” is algorithm 1? “ADMMI” is algorithm 2? Where is the thinning operation in the description of these algorithms?
I understood that the ADMMI algorithm was proposed by the authors of the paper. But where did it come from? You need either a link to the authors' work where they have already justified the derivation of Algorithm 2, or a brief description of how Algorithm 2 was derived. For me it is not obvious at all!!!
The number of figures can be significantly reduced. I ran through them with my eyes and practically did not compare them with each other. All the information is in the tables.
Between figs. 4, 5, 6 and 7 the difference is hardly noticeable. Is this not a mistake?
In Table 1, the rows “ADMMO” and “ADMMI” have the same numbers. Is this a misprint?
I did not understand the following: in Figs. 8 and 9 at K = 5 the noise is less than in Figs. 9 and 10 at K = 20. At K = 20 the sparsity is smaller and the reconstructed images should have less noise. I assume so! Similar question to the figures showing the images of model 3.
Author Response
Reviewer 4:
The paper is devoted to the actual problem of reducing the time of registration of bistatic radar echoes when operating in sparse mode, but with image acquisition that could be obtained by careful scanning of space.
Respond: Thank for the detailed comments, and we check the whole paper and modified according to the comments below. Thanks again for the time and efforts in helping us to improve our manuscript.
Comment1: I recommend a figure at the very beginning of Section 2 with a brief explanation of the principle of operation and an indication of the variables used.
Respond1: We add a Figure to the beginning of Section 2 to give a brief explanation of the principle of operation.(line 86-line 89)
Comment2: It is necessary to explain why the noise in the image pixels is Laplace distributed rather than Rayleigh distributed, which is typical for images. At least optical ones. Or make a proper reference!
Respond2: Thanks very much for the detailed comment. In the revised manuscript, the reference(reference [13]) of the assumption of noise is added.(line 114)
Comment3: In formula (11), what is F F=2? Or does the explanation only appear in formula (13)? And how does formula (13) differ from the usual l2 norm?
Respond3: Thank you very much for the kindly advise. We checked the use of ||.||F and ||.||2 in the paper, and we are very sorry for the confusion using of the notation of norm. Actually, ||.||2 is used for the vector which is defined as , and ||.||F is used for the matrix which is defined as , the explanation is added in the revised paper (line 134-line136). Thanks very much again for the time in revising our paper.
Comment4: In section 4, we need to establish a correspondence between the numbers 34 and 128 and the variables M, Q, N and K. How were the images in Figure 1 modeled. Is this image X? What is the value of K for these images?
Respond4: Thank you very much for the comments. We give an explanation numbers 34 and 128 and the variables mentioned in this comment.(line 190-line 195)
Comment5: In Fig. 1, the numbers of samples are plotted on the axes. We need to specify the physical quantities on both axes as well.
Respond5: We specify the physical quantities on both axes.(line 190-line 191), the physical quantity of the axes is the number of the image pixels.
Comment6: I don't understand what “three measurement positions of ground 10, 30, and 60” means. I need an explanation.
Respond6: We are very sorry for the confusion. The intended meaning of this statement refers to the 10th, 30th, and 60th measurement instances. The translation error has been corrected (Line 207).
Comment7: “ADMMO” is algorithm 1? “ADMMI” is algorithm 2? Where is the thinning operation in the description of these algorithms?
Respond7: We are very sorry for the confusion. We specify the difference between ADMM and our proposal. Algorithm 1 is the original ADMM with fixed multiplier and iterative variables V. And in Algorithm 2, the multiplier and iterative variable is updated according to the iterations. Thus the proposed algorithm is the combination of Algorithm 1 and Algorithm 2. We indicated our proposal in the revised paper to avoid misunderstandings. (line 161)
Comment8: I understood that the ADMMI algorithm was proposed by the authors of the paper. But where did it come from? You need either a link to the authors' work where they have already justified the derivation of Algorithm 2, or a brief description of how Algorithm 2 was derived. For me it is not obvious at all!!!
Respond8: Thank you very much for the comments. The consideration of our proposal is added to the revised paper. (line 175-line 186).
Comment9: The number of figures can be significantly reduced. I ran through them with my eyes and practically did not compare them with each other. All the information is in the tables.
Respond9: Thanks very much for all the time spent in the revision. We delete several figures (Figure 6, Figure 7) which do not contain much information in the revised paper. Thanks again for the advices.
Comment10: Between figs. 4, 5, 6 and 7 the difference is hardly noticeable. Is this not a mistake?
Respond10: Thanks very much for the comments. We marked the difference area in Figure 4 and Figure 5 by the white rectangles(line 241, line 244). And we delete Figure 6 and Figure 7 by taking into account the above comment.
Comment11: In Table 1, the rows “ADMMO” and “ADMMI” have the same numbers. Is this a misprint?
Respond11: Thanks very much for the detailed comment. We checked again the results in this experiment, the original ADMM and our proposal get the same RRF, thus the same numbers are shown in Table 1 for Model 1.
Comment12: I did not understand the following: in Figs. 8 and 9 at K = 5 the noise is less than in Figs. 9 and 10 at K = 20. At K = 20 the sparsity is smaller and the reconstructed images should have less noise. I assume so! Similar question to the figures showing the images of model 3.
Respond12: We are sorry for not explaining the results clearly. The reason is added to the paper (line 326-line 330). The sparsity balances the fitting error and the suppression of the noise. If small sparsity K is used, the noise is suppressed as well as the original signal, so the fitting error between the reconstructed image and the original image is large, but the reconstructed images have less noise. If large sparsity K is used, the image will be recovered much better as well as the noise. So the reconstructed images have much more noise than recovered results with small sparsity K. Thanks very much for this comments.

Round 2
Reviewer 1 Report
Comments and Suggestions for Authors
All questions have been answered.
The manuscript has been sufficiently improved to warrant publication in Sensors.
Comments on the Quality of English LanguageThere are still some syntax issues, please check in detail and revise.
Reviewer 2 Report
Comments and Suggestions for Authors
the authors have made the necessary revision and it is ready for publishing.
Reviewer 3 Report
Comments and Suggestions for Authors
- No comments.
Comments on the Quality of English Language- The quality of English can be improved.